# Postnatal Exposure to the Endocrine Disruptor Dichlorodiphenyltrichloroethane Affects Adrenomedullary Chromaffin Cell Physiology and Alters the Balance of Mechanisms Underlying Cell Renewal

**DOI:** 10.3390/ijms25031494

**Published:** 2024-01-25

**Authors:** Nataliya V. Yaglova, Sergey S. Obernikhin, Svetlana V. Nazimova, Dibakhan A. Tsomartova, Ekaterina P. Timokhina, Valentin V. Yaglov, Elina S. Tsomartova, Elizaveta V. Chereshneva, Marina Y. Ivanova, Tatiana A. Lomanovskaya

**Affiliations:** 1Laboratory of Endocrine System Development, A.P. Avtsyn Research Institute of Human Morphology of Federal State Budgetary Scientific Institution “Petrovsky National Research Centre of Surgery”, 119991 Moscow, Russia; lesd@morfolhum.ru (S.S.O.); pimka60@list.ru (S.V.N.); dtsomartova@mail.ru (D.A.T.); rodich@mail.ru (E.P.T.); vyaglov@mail.ru (V.V.Y.); tselso@yandex.ru (E.S.T.); 2Department of Human Anatomy and Histology, Federal State Funded Educational Unlike the Control Institution of Higher Education, I.M. Sechenov First Moscow State Medical University, 119435 Moscow, Russia; yelizaveta.new@mail.ru (E.V.C.); ivanova_m_y@mail.ru (M.Y.I.); tatiana_80_80@inbox.ru (T.A.L.)

**Keywords:** adrenal medulla, DDT, endocrine disrupting chemicals, postnatal exposure, growth, self-renewal, proliferation, progenitor cells

## Abstract

Dichlorodiphenyltrichloroethane (DDT) is a wide-spread systemic pollutant with endocrine disrupting properties. Prenatal exposure to low doses of DDT has been shown to affect adrenal medulla growth and function. The role of postnatal exposure to DDT in developmental disorders remains unclear. The aim of the present investigation is to assess growth parameters and the expression of factors mediating the function and renewal of chromaffin cells in the adult adrenal medulla of male Wistar rats exposed to the endocrine disruptor o,p’-DDT since birth until sexual maturation. The DDT-exposed rats exhibited normal growth of the adrenal medulla but significantly decreased tyrosine hydroxylase production by chromaffin cells during postnatal period. Unlike the control, the exposed rats showed enhanced proliferation and reduced expression of nuclear β-catenin, transcription factor Oct4, and ligand of Sonic hedgehog after termination of the adrenal growth period. No expression of pluripotency marker Sox2 and absence of Ascl 1-positive progenitors were found in the adrenal medulla during postnatal ontogeny of the exposed and the control rats. The present findings indicate that an increase in proliferative activity and inhibition of the formation of reserve for chromaffin cell renewal, two main mechanisms for cell maintenance in adrenal medulla, in the adult DDT-exposed rats may reflect a compensatory reaction aimed at the restoration of catecholamine production levels. The increased proliferation of chromaffin cells in adults suggests excessive growth of the adrenal medulla. Thus, postnatal exposure to DDT alters cell physiology and increases the risk of functional insufficiency and hyperplasia of the adrenal medulla.

## 1. Introduction

Exposure to endocrine disrupting chemicals is a global problem for humans and animals. Endocrine disrupting chemicals have been defined by the United States Environmental Protection Agency (EPA) as “an agent that interferes with the synthesis, secretion, transport, binding, or elimination of natural hormones in the body that are responsible for the maintenance of homeostasis, reproduction, development and/or behavior” [1]. Exposure to endocrine disrupting chemicals usually begins in utero since most of them are lipophilic substances with low molecular weight. This allows endocrine disruptors to penetrate blood–tissue barriers including the placental barrier [2,3,4]. The time of exposure to disruptors is a stumbling block for determining the consequences of their action in a developing organism. First of all, this is due to the fact that, having begun in the prenatal period, the impact of endocrine disruptors continues after birth. The prenatal period of ontogeny is considered the most vulnerable because of the double negative outcomes of exposure affecting developing fetal organs directly and indirectly through disorders produced by endocrine disruptors in maternal organisms. Interference in the differentiation of the ectoderma, endoderma, and mesoderma and the further formation of organs is known to provoke developmental abnormalities and alterations in the realization of the developmental program [5,6,7,8]. Affection of the proper proliferation, differentiation, and migration of cells underlies the risks of oncological diseases and insufficiency in organ function. The impact of postnatal exposure on morphological and physiological changes induced by endocrine disruptors is harder to assess since it usually follows prenatal exposure [9,10,11,12]. It is not clear whether postnatal exposure aggravates disorders produced by prenatal endocrine disruption or initiates additional changes in developing organisms.

Among endocrine disrupting chemicals, the most wide-spread substances are persistent organochlorine pesticides [13,14]. Organochlorinated pesticides comprise dozens of chemicals actively used for the protection of crops and in vector-borne disease control [15]. Dichlorodiphenyltrichloroethane (DDT) was the first insecticide used worldwide that was later found to have endocrine disrupting properties [13,16]. It is still recommended by the World Health Organization and used in many countries as an effective tool to kill malaria vectors [17]. DDT and its metabolites have a long half-life and persist in soils and water, providing continuous exposure for the population [18,19,20]. DDT easily penetrates the placental barrier and can occur in even higher concentrations since pregnancy alters the kinetics and metabolism of some endocrine disrupting chemicals, including DDT, resulting in its more intense accumulation compared to non-pregnant females [2,4]. Recent investigations have shown that DDT interferes in thyroid, adrenal, and testicular and ovary functioning [21,22,23,24,25,26].

In our previous investigations, we found that low-dose exposure to DDT significantly reduces catecholamine production by adrenal chromaffin cells by the inhibition of tyrosine hydroxylase [27,28,29]. Further research has shown that prenatal and postnatal onset of exposure resulted in a similar decrease in epinephrine and norepinephrine synthesis. The chromaffin cells also exhibited a diminished number of mitochondria, especially in subplasmalemmal regions mediating the secretion of catecholamine-containing granules [28,29]. Continuous prenatal and postnatal exposure to DDT has also been found to slow down the growth of the adrenal medulla and affect the formation of the pool for the self-renewal of chromaffin cells [30]. The role of postnatal exposure to DDT in developmental disorders remains unclear. In the present investigation, we focused on an assessment of growth parameters and the expression of factors mediating the maintenance of chromaffin cells in the adult adrenal medulla of rats exposed to the endocrine disruptor DDT since birth.

## 2. Results

### 2.1. Histology and Histomorphometry of the Adrenal Medulla

On the 42nd day of postnatal development, the control rats exhibited well-developed adrenal medulla with clusters of basophilic cells separated by capillaries and venules (Figure 1A). The postnatally exposed rats had no significant difference in adrenal medulla histology (Figure 1B). Histomorphometry showed that the size of the adrenal medulla and the portion of the area of chromaffin cells in the surface area in the exposed animals were similar to the control values (Figure 1E,F).

On the 70th day of postnatal development, the control rats showed an increase in the size of the adrenal medulla compared to the 42nd day. The portion of the area of chromaffin cells did not change with age. The chromaffin cells exhibited less basophilic cytoplasm and enlightened nuclei (Figure 1C–E). The size of the adrenal medulla in the postnatally exposed rats did not differ significantly from that in the control. The portion of the area of chromaffin cells was also within the control values. No significant changes in the structure of chromaffin cells were observed (Figure 1D–F).

### 2.2. Tyrosine Hydroxylase Content in the Adrenomedullary Chromaffin Cells

The control rats exhibited an extremely high content of tyrosine hydroxylase in the cytoplasm of all chromaffin cells, which did not change from the 42nd to the 70th day of postnatal development (Figure 2A,C, Table 1).

The postnatally exposed rats exhibited a significantly lower tyrosine hydroxylase content on both the 42nd and the 70th day (Figure 2B,D, Table 1).

### 2.3. Proliferation Rate of the Adrenomedullary Chromaffin Cells

Evaluation of Ki-67 expression in the control rats revealed the downregulation of chromaffin cell proliferation on the 70th day of postnatal development (Figure 3A,C,E). The postnatally exposed rats had a lower rate of proliferation on the 42nd day and surprisingly accelerated chromaffin cell division on the 70th day of postnatal development (Figure 3B,D,E).

### 2.4. Oct4 Expression

Oct4-positive chromaffin cells were observed in the adrenal medulla of the control rats at both ages (Figure 4A,C). It is noteworthy that their percentage on the 70th day was three times higher than that on the 42nd day of postnatal development (Figure 4E).

Evaluation of Oct4 expression in the DDT-exposed rats also revealed an increase in the percentage of positive adrenal chromaffin cells with age, but it was lower than that in the control because of the higher content of Oct4-expressing chromaffin cells on the 42nd day and the lower value on the 70th day of postnatal development (Figure 4B,D,E).

### 2.5. Sox2 Expression

Immunohistochemical detection of transcription factor Sox2 did not reveal positive cells in the control and DDT-exposed rats during postnatal development (Figure 5).

### 2.6. Ascl1 Expression

Ascl1-positive cells were not found in the adrenal medulla of the rats postnatally exposed to DDT or in the control (Figure 6).

### 2.7. Sonic Hedgehog (Shh) Ligand Expression

The control rats exhibited an extremely low content of Shh-positive chromaffin cells in the adrenal medulla on the 42nd day and a significant increase in their number on the 70th day (Figure 7A,C,E). Unlike the control, the exposed rats had a higher percentage of Shh-positive chromaffin cells on the 42nd day and a smaller increase by the 70th day (Figure 7B,D,E).

### 2.8. Wnt Signaling Activation

Quantification of chromaffin cells with β-catenin translocated into the nucleus on the 42nd and 70th days revealed the upregulation of Wnt signaling activation in the adrenal medulla of the control rats (Figure 8A,C,E). The DDT-exposed rats demonstrated another pattern of Wnt signaling activation. The percentage of chromaffin cells with β-catenin-positive nuclei was similar to the control values on the 42nd day and showed no increase with age (Figure 8B,D,E).

## 3. Discussion

The data obtained showed that low-dose postnatal exposure since birth did not affect the postnatal growth of the adrenal medulla. Our previous investigation found a delayed decline in the growth rates of the adrenal medulla after puberty in rats prenatally and postnatally exposed to DDT [30]. This is most likely due to shifts in the growth and development program of the adrenals. The present findings clearly showed that DDT did not disrupt the growth parameters of the adrenal medulla up to the 70th day of the postnatal period when the adrenal glands reached their maximal development. We observed no principal differences in adrenal medulla histology in the exposed rats, but an evaluation of proliferation rates and the expression of transcription factors regulating cell division and differentiation showed that the growth process of the organ differed from the control pattern. Lower rates of proliferation found on the 42nd day may reflect previous accelerated growth since the histological parameters (surface area of the adrenal medulla and portion of chromaffin tissue in the medulla) indicated that the development of the adrenal medulla was similar to control parameters. This means that prior to the 42nd day, the proliferation rate of chromaffin cells was adequate or accelerated after possible attenuation earlier in the postnatal period. Surprisingly, the adrenal chromaffin cells of rats postnatally exposed to DDT exhibited accelerated proliferation on the 70th day. Increased proliferation suggests further growth of the adrenal medulla. We propose that proliferation was stimulated by secretory insufficiency of the chromaffin cells. The significantly reduced content of tyrosine hydroxylase, a key enzyme in catecholamine synthesis, found at both ages, may allow compensatory division of chromaffin cells.

Some investigations of the nervous tissue and adrenal medulla regeneration and chromaffin cell turnover during postnatal ontogeny show that the inhibition of growth triggers additional proliferation mechanisms to maintain the chromaffin cell population [31,32,33]. The control rats showed an increase in the number of cells expressing morphogens and pluripotency factors like Shh and Oct4 and cells with activated canonical Wnt signaling. It is noteworthy that all these cells were chromaffin cells since all of them produced tyrosine hydroxylase and had typical cytomorphology. The activation of canonical Wnt signaling together with the higher expression of the pluripotency factor Oct4 and morphogen Shh ligand is indicative of an increased pool of the cells capable of dedifferentiation and further proliferation to maintain tissue homeostasis during postnatal life [33].

The present investigation revealed the down-regulation of tyrosine hydroxylase expression by chromaffin cells in rats postnatally exposed to low doses of DDT. We observed chromaffin cells with both a low content tyrosine hydroxylase and tyrosine hydroxylase-negative cytoplasm. The negative cells were found in pubertal and postpubertal periods. These data suggest a dual interpretation. On the one hand, the absence of tyrosine hydroxylase may result from a strong inhibition of its synthesis by the endocrine disruptor, but on the other hand, this may indicate low differentiation of cells. The presence of few undifferentiated progenitors in the adult adrenal medulla is a controversial issue. Earlier studies showed that cell maintenance in the adrenal medulla is attributed to the proliferation of chromaffin cells [34,35]. Later investigations demonstrated a presence of progenitor cells in the postnatal adrenal medulla with the upregulated expression of neuronal (nestin, vimentin, musashi 1) and even neural crest (Sox1, Sox9) markers [36,37,38]. In the present study, we performed an immunohistochemical evaluation of Ascl1, a protein that plays a role in the neuronal commitment and differentiation of Schwann cell and chromaffin cell precursors and is also expressed in neuroendocrine tumor cells during the generation of olfactory and autonomic neurons [39,40]. Transcription factor Ascl1 has been also found to regulate the cell cycle as well as canonical Wnt signaling [41,42]. In our study, we did not identify cells with immunohistochemically detectable levels of the Ascl1 protein in the control group or in rats exposed to the endocrine disrupter DDT during any of the studied age periods. The results obtained allow us to exclude the appearance of neuronal precursors in the adrenal medulla of the DDT-exposed rats.

We also evaluated the expressions of two additional transcription factors, Sox2 and Oct4. Both are associated with the maintenance of the pluripotent state of cells and cell de- and transdifferentiation [43,44]. Transcription factor Sox2 has been shown to mediate the renewal of adrenal chromaffin cells, and SOX2-expressing cells of the adult adrenal medulla have been considered true stem progenitors of catecholamine-secreting chromaffin cells [45]. In our investigation, immunohistochemical detection did not reveal Sox2-positive cells, whereas Oct4 expression was found at both ages. Postnatal exposure to DDT did not change the age-related dynamics during the expression of Oct4. However, the percentage of Oct4-positive chromaffin cells was higher in the pubertal period and lower than that in the control after puberty. The same changes were observed in Shh expression. The Shh pathway has been shown to provide reparation of nervous tissue in postnatal ontogeny and is considered a pathway that allows reprogramming of cells [31,32,46]. The results indicated that higher numbers of Oct4- and Shh-positive cells in the adrenal medulla were associated with the inhibition of proliferation, and a lower increase in the number of Oct4- and Shh-expressing cells was in parallel with the activation of cell division.

Unlike Oct4 and Shh, an evaluation of canonical β-catenin/Wnt signaling activation in DDT-exposed rats revealed minor changes. Canonical Wnt signaling is a generally recognized pathway that regulates cell division, differentiation, reprogramming, and survival [47]. Earlier reports have demonstrated an implication of canonical Wnt signaling in the maintenance of various cell populations [48,49,50,51]. The increase in proliferative activity observed on the 70th day of postnatal life was associated with the inhibition of β-catenin translocation into nuclei. Taken together, all of the findings demonstrate that postnatal exposure to DDT shifts the balance of proliferation and the formation of the reserve pool of cells in adult rats.

## 4. Materials and Methods

### 4.1. Animals

Adult male and female Wistar rats of the same age were obtained from the Scientific Center of Biomedical Technologies of the Federal Medical and Biological Agency of Russia. All of the rats were housed in the local vivarium with 5 animals per cage. The housing conditions were as follows: temperature +22–23 °C, humidity 60–65%, and 12/12 h light/darkness cycle. The rats were given a pelleted standard chow. The access to food and water was free. The investigation was performed in accordance with the handling standards and rules of laboratory animals, consistent with the “International Guidelines for Biomedical Researches with Animals” (1985), in accordance with GOST 33215-2014 (Guidelines for Accommodation and Care of Animals. Environment, Housing and Management), and GOST 33216-2014 (Guidelines for Accommodation and Care of Animals. Species-Specific Provisions for Laboratory Rodents and Rabbits), as well as routine laboratory standards in the Russian Federation (Order of Ministry of Healthcare of the Russian Federation dated 19 June 2003 No. 267) Animal procedures were approved by the Ethics committee of the Research Institute of human morphology (protocol No 28(4), 28 October 2021).

### 4.2. Experimental Design

The female rats (weight: 180–220 g) received a solution of 20 µg/L o,p-DDT (Sigma-Aldrich, Saint Louis, MO, USA) in tap water ad libitum after parturition during lactation. The o,p-isomer of DDT was preferred due its higher solubility in water (80 µg/L) [52]. After weaning (3 weeks of age), the progeny of the rat dams received the same solution of o,p-DDT during postnatal development up to the 70th day of age when rat adrenals reach their maximal development [53]. Only male progeny was enrolled for examination (n = 20) to avoid changes in some functional parameters of the adrenals evoked by fluctuations of female sex hormones during the ovarian cycle. Male progeny of intact female rats was used as a control (n = 20). Half of the rats were sacrificed in the pubertal period at the age of 42 days, and their adrenals were collected, whereas those of the others were collected after puberty at the age of 70 days by zoletil overdosage. The rats’ body weight and amount of consumed water were measured daily to calculate the daily intake of DDT. The exposed and the control progeny had no differences in body weight during postnatal development. The average daily intake of DDT by the dams during lactation was 2.69 ± 0.18 µg/kg bw, and by progeny after weaning—2.98 ± 0.13 µg/kg bw, which corresponded to DDT consumption by humans through food products with consideration for differences in DDT metabolism between rats and humans [53,54]. The absence of DDT, its metabolites, and related organochlorine compounds in the tap water and chow was confirmed by gas chromatography at the Moscow Federal Budgetary Institution of Public Health.

### 4.3. Adrenal Histology

The removed adrenal glands were fixed in Bouin solution. After standard histological processing, the tissue samples were embedded in paraffin. Equatorial sections of the adrenals were stained with hematoxylin and eosin. Histological examination was performed with a Leica DM2500 light microscope (Leica Microsystems Gmbh, Wetzlar, Germany). Computer histomorphometry of the light microscope images was carried out using ImageScope software version M (Leica Microsystems Gmbh, Wetzlar, Germany). The size of the adrenal medulla and the portion of the area of chromaffin cells were measured.

### 4.4. Immunohistochemistry

Immunohistochemical evaluation of Ki-67, tyrosine hydroxylase, Sox2, Ascl1, Oct4, β-catenin, and Hedgehog (Shh) was performed using paraffin-embedded tissues. After antigen retrieval with 10 mM sodium citrate (pH 6.0), endogenous peroxidase and endogenous immunoglobulins were blocked with Hydrogen Peroxide Block and Protein Block (Thermo Fisher Scientific, Waltham, MA, USA). The slides were incubated with primary antibodies to Ki-67 (1:100, Cell Marque, Rocklin, CA, USA), tyrosine hydroxylase (1:1000, Abcam, Cambridge, MA, USA), Sox2 and Ascl1 (from 1:400 to 1:50), β-catenin (1:100, Cell Marque, Roklin, CA, USA), Shh (1:400, Abcam, Cambridge, MA, USA), and Oct4 (1:5000, Abcam, Cambridge, MA, USA) overnight at 8ºC. Sections of rat embryonic tissues were used as a positive control for Oct4, Sox2, and Ascl1. Slides processed without incubation with primary antibodies were used as a negative control. The reaction was visualized with the UltraVision LP Detection System reagent kit (Thermo Fisher Scientific, Waltham, MA, USA) according to the manufacturer’s recommendations. The sections were counterstained with Mayer’s hematoxylin.

Activation of canonical Wnt signaling was assessed by the percentage of chromaffin cells with β-catenin positive nuclei [55]. The chromaffin cells were examined at a magnification of 1000 to detect negative cells and cells with diffuse and local distribution of tyrosine hydroxylase. The percentage of tyrosine hydroxylase-positive types and tyrosine hydroxylase-negative chromaffin cells was calculated. The expressions of Ki-67, Shh, and Oct4 were assessed as a percentage of immunopositive cells with nuclear staining.

### 4.5. Statistical Analysis

The statistical analyses were carried out using the software package Statistica 7.0 (StatSoft, Tulsa, OK, USA). The central tendency and dispersion of quantitative traits with approximately normal distribution were presented as the mean and standard error of the mean (M ± SEM). Quantitative comparisons of independent groups were performed using Student’s *t*-test taking into account the values of Levene’s test for the equality of variances. Quantitative comparisons were performed using Chi-square. Differences were considered statistically significant at *p* < 0.05.

## 5. Conclusions

Postnatal exposure to low doses of DDT does not change the parameters of adrenal medulla growth but significantly suppresses the synthesis of tyrosine hydroxylase in chromaffin cells. The evoked imbalance of the two main mechanisms of cell turnover, that is, the increase in the proliferative activity of adrenomedullary cells and the inhibition of the formation of reserve for chromaffin cell renewal in the adult rats, may reflect a compensatory reaction aimed at the restoration of catecholamine production levels. The increased proliferation of chromaffin cells in adults suggests excessive growth of the adrenal medulla, which may result in hyperplastic or even neoplastic transformation and secretory disorders, especially in the case of an imbalance in cell proliferation and total secretory activity. The data obtained allow us to consider postnatal exposure to DDT as a risk factor of adrenomedullary insufficiency and hyperplasia. The results of the investigation clearly demonstrate that negative outcomes of postnatal exposure to the endocrine disruptor DDT are associated with the interference of catecholamine synthesis and the transcriptional control of the regenerative potential of chromaffin cells.

## Figures and Tables

**Figure 1 ijms-25-01494-f001:**
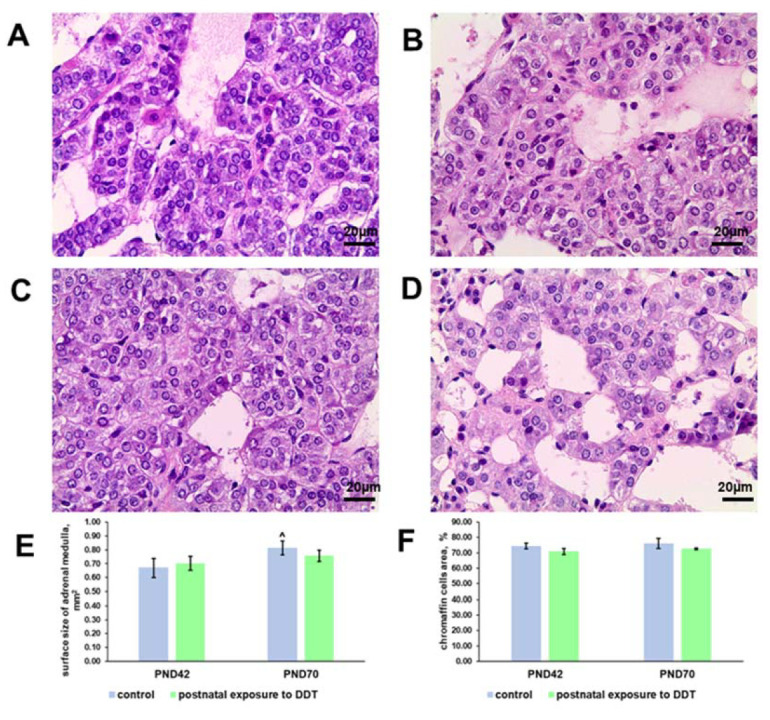
Changes in the histology and histomorphometry of the adrenal medulla in rats exposed postnatally to dichlorodiphenyltrichloroethane (DDT). Histology of the adrenal medulla of the control rats on the 42nd (**A**) and 70th (**C**) day of postnatal development and of the rats postnatally exposed to DDT on the 42nd (**B**) and 70th (**D**) day. Magnification 400. Surface area of the adrenal medulla (**E**), total chromaffin cell area (**F**). Data shown are the mean ± S.E.M., *p* < 0.05 compared to the 42nd day (^).

**Figure 2 ijms-25-01494-f002:**
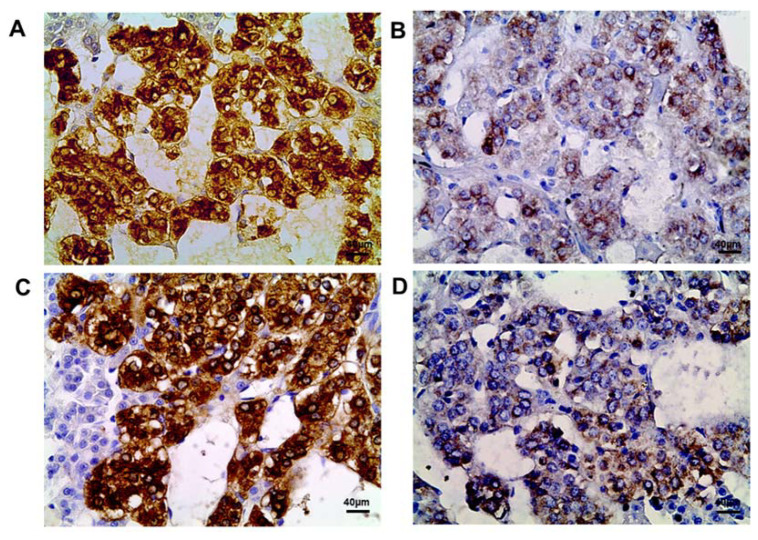
Immunohistochemical detection of tyrosine hydroxylase in the control rats on the 42nd (**A**) and 70th (**C**) day of postnatal development and in rats postnatally exposed to DDT on the 42nd (**B**) and 70th (**D**) day of postnatal development, magnification 400.

**Figure 3 ijms-25-01494-f003:**
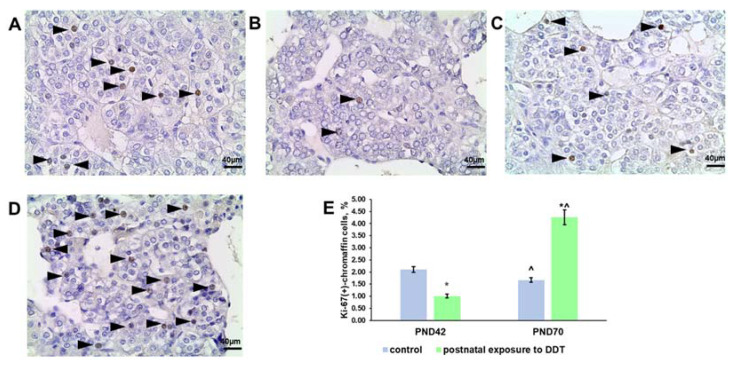
Immunohistochemical detection of Ki-67 in the adrenal medulla of the control rats on the 42nd (**A**) and 70th (**C**) day of postnatal development and of rats postnatally exposed to DDT on the 42nd (**B**) and 70th (**D**) day of postnatal development, magnification 400. Immunopositive cells are indicated with arrowheads; percentage of Ki-67-positive chromaffin cells in the adrenal medulla (**E**). Data shown are the mean ± S.E.M., * *p* < 0.05 compared to the control of the appropriate age (*), compared to the 42nd day (^).

**Figure 4 ijms-25-01494-f004:**
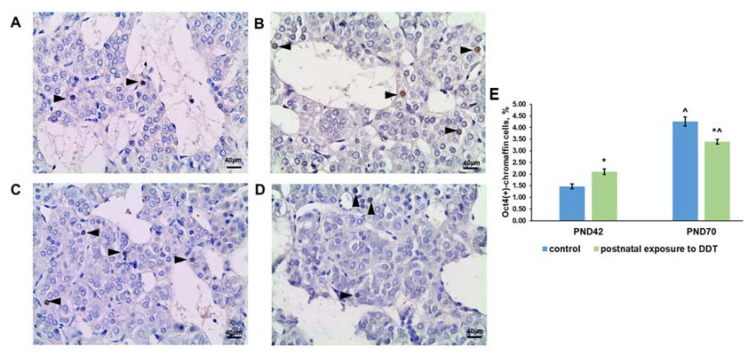
Immunohistochemical detection of transcription factor Oct4 in the adrenal medulla of the control rats on the 42nd (**A**) and 70th (**C**) day of postnatal development and of rats postnatally exposed to DDT on the 42nd (**B**) and 70th (**D**) day of postnatal development, magnification 400. Immunopositive cells are indicated with arrowheads; percentage of Oct4-positive chromaffin cells in the adrenal medulla (**E**). Data shown are the mean ± S.E.M., * *p* < 0.05 compared to the control of the appropriate age (*), compared to the 42nd day (^).

**Figure 5 ijms-25-01494-f005:**
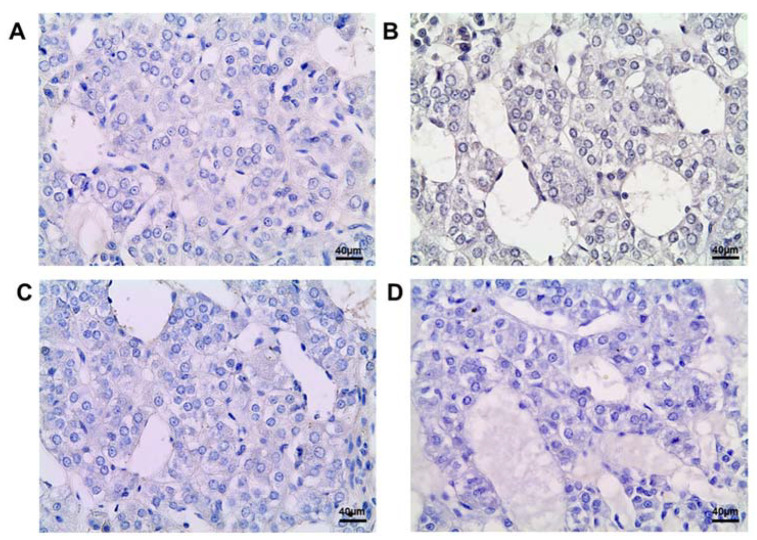
Immunohistochemical detection of transcription factor Sox2 in the adrenal medulla of the control rats on the 42nd (**A**) and 70th (**C**) day of postnatal development and of rats postnatally exposed to DDT on the 42nd (**B**) and 70th (**D**) day of postnatal development, magnification 400.

**Figure 6 ijms-25-01494-f006:**
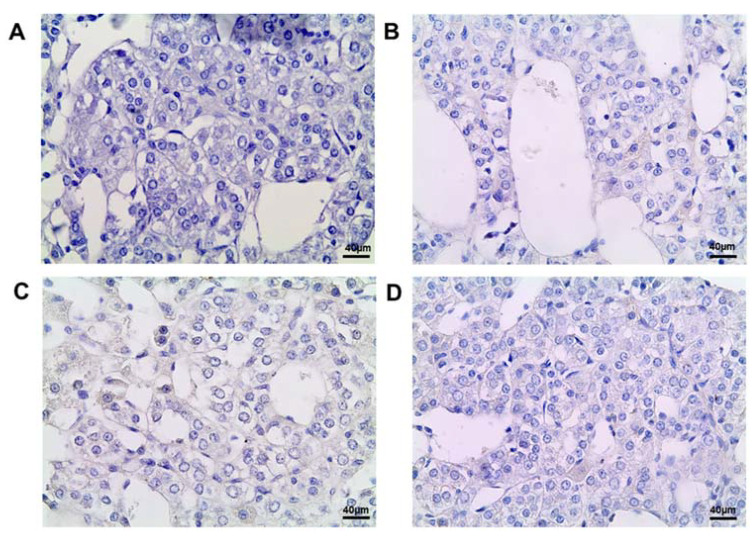
Immunohistochemical detection of transcription factor Ascl1 in the adrenal medulla of the control rats on the 42nd (**A**) and 70th (**C**) day of postnatal development and of rats postnatally exposed to DDT on the 42nd (**B**) and 70th (**D**) day of postnatal development, magnification 400.

**Figure 7 ijms-25-01494-f007:**
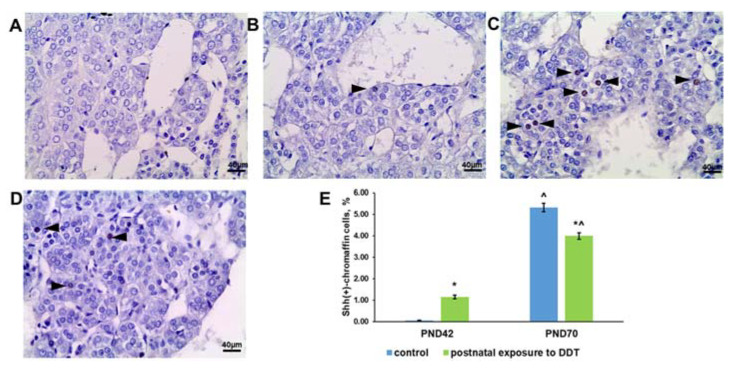
Immunohistochemical detection of Sonic Hedgehog ligand in the adrenal medulla of the control rats on the 42nd (**A**) and 70th (**C**) day of postnatal development and of rats postnatally exposed to DDT on the 42nd (**B**) and 70th (**D**) day of postnatal development, magnification 400. Immunopositive cells are indicated with arrowheads; percentage of Shh-positive chromaffin cells in the adrenal medulla (**E**). Data shown are the mean ± S.E.M., * *p* < 0.05 compared to the control of the appropriate age (*), compared to the 42nd day (^).

**Figure 8 ijms-25-01494-f008:**
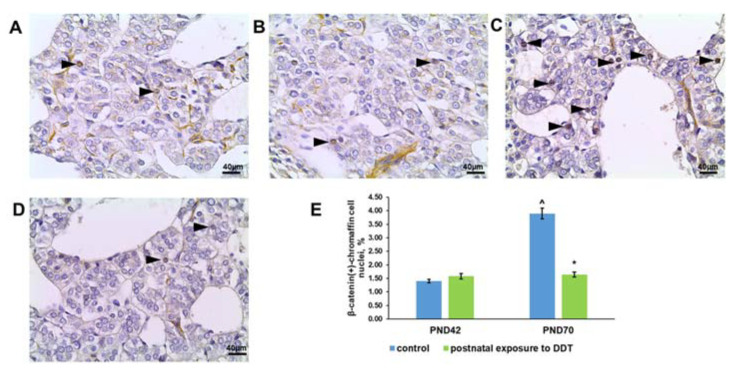
Immunohistochemical detection of β-catenin in the adrenal medulla of the control rats on the 42nd (**A**) and 70th (**C**) day of postnatal development and of rats postnatally exposed to DDT on the 42nd (**B**) and 70th (**D**) day of postnatal development, magnification 400. Immunopositive cells are indicated with arrowheads; percentage of chromaffin cells with β-catenin translocated into nuclei in the adrenal medulla (**E**). Data shown are the mean ± S.E.M., * *p* < 0.05 compared to the control of the appropriate age (*), compared to the 42nd day (^).

**Table 1 ijms-25-01494-t001:** Percentages of adrenal chromaffin cells with different tyrosine hydroxylase (TH) contents in the cytoplasm in the control rats and rats postnatally exposed to dichlorodiphenyltrichloroethane (DDT) at the age of 42 and 70 days.

	42nd Day		70th Day	
	Control Group	Postnatal Exposure to DDT	Control Group	Postnatal Exposure to DDT
Chromaffin cells with diffuse distribution of TH in cytoplasm	93.20 ± 2.16	12.75 ± 4.89 *	85.30 ± 4.01	11.69 ± 2.79 *
Chromaffin cells with local distribution of TH in cytoplasm	6.80 ± 2.14	59.18 ± 9.13 *	14.70 ± 0.92	64.64 ± 3.24 *
TH-negative cells	0	28.1 ± 6.04 *	0	23.67 ± 2.92 *

Data shown are the mean ± S.E.M. * *p* < 0.05 compared to the control of the appropriate age.

## Data Availability

The data presented in this study are available from the corresponding author upon reasonable request.

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
