# Peer review of "Postnatal Exposure to the Endocrine Disruptor Dichlorodiphenyltrichloroethane Affects Adrenomedullary Chromaffin Cell Physiology and Alters the Balance of Mechanisms Underlying Cell Renewal"

_ijms, 2024, doi:10.3390/ijms25031494_

Round 1

Reviewer 1 Report

Comments and Suggestions for Authors

The article by Yaglova and co-authors is devoted to the study of the postnatal effects of the endocrine disruptor DDT on the physiology of adrenomedullary chromaffin cells and the balance of the mechanisms underlying cell renewal.

The introduction substantiates the relevance of the work. Literary sources on this topic are given.

The experimental part presents microphotographs and distribution histograms. The quality of histogram presentation could be improved. When enlarged, the font blurs.

In the discussion section, a competent analysis of the results obtained is carried out, as well as a comparison with known scientific data.

The conclusion section requires improvement. It is very short and non-specific. Perhaps the authors will be able to transfer some of the conclusions here from the discussion section.

In the materials and methods section, the authors indicate that the animal studies were conducted in accordance with “The investigation was performed in accordance with the handling standards and 280 rules of laboratory animals as consistent with “International Guidelines for Biomedical 281 Re-searches with Animals” ( 1985), laboratory routine standards in the Russian Federation- 282 tion (Order of Ministry of Healthcare of the Russian Federation dated 06/19/2003 No.267) 283 and “Animal Cruelty Protection Act” dated 12/1/1999, regulations of experimental ani- 284 mal operation approved by Order of Ministry of Healthcare of USSR No.577 dated 285 08/12/1977.”

In my opinion these are very old protocols. Perhaps modern ones should be specified.

In general, the work of Yaglova and co-authors is of undoubted scientific interest and, after minor modifications, can be accepted for publication.

Author Response

Dear Reviewer! Thank you for your appreciation of our work.

We updated the regulations and added that the experiment was conducted in accordance with GOST 33215-2014 (Guidelines for Accommoda­tion and Care of Animals. Environment, Housing and Management) and GOST 33216-2014 (Guidelines for Accommodation and Care of Animals. Species-Specific Provisions for Laboratory Rodents and Rabbits).

Reviewer 2 Report

Comments and Suggestions for Authors

The manuscript “Postnatal exposure to the endocrine disruptor DDT affects adrenomedullary chromaffin cells physiology and alters balance of mechanisms underlying cell renewal” reports a very interesting and important topic, exploring the effects of postnatal exposure to DDT on the adrenal medulla growth and function. Some suggestions to improve clarity and add context follows, but overall, this manuscript is well structured with interesting results.

Comments:

1.       Is the tested concentration of DDT considered realistic? What were the assumptions to choose it? And why did the author only analysed one dose of DDT, considering the endocrine disrupting action of this compound?

2.       Line 296: what do the authors mean by “intact female rats”?

3.       Lines 303-304: this sentence is very confusing. What are the DDT consumption values for humans? Please explain this.

4.       When were the adrenal glands removed, at the 2 timepoints (42 and 70 days)? It is not very clear, maybe a scheme of the methods would be helpful.

5.       Line 325: why was embryonic tissue used as positive control?

6.       Lines 202-206: can you explain these differences, between the previous study and the present one?

7.       Line 210: please explain and elaborate this sentence.

8.       Line 353: how excessive can this growth be and what are the consequences? Please, further explore this issue.

9.       In the conclusion it should be explained how these results can contribute to new knowledge in the scientific area. How can the authors extrapolate these results to the human being? Compare DDT human exposure values to those used in the study.

10.   Please propose a mechanism for the results and conclusions obtained in this manuscript (10.1016/j.mam.2021.101054, 10.1016/j.envpol.2023.121366 and 10.3389/fendo.2023.1059854).

Overall, the manuscript is well written with a simple and understandable English, with some minor spealing mistakes that could be corrected.

Author Response

Dear reviewer! Thank you very much for your recommendation!

We corrected the manuscript according to your comments.

Comments and responses:

  1. Is the tested concentration of DDT considered realistic? What were the assumptions to choose it? And why did the author only analysed one dose of DDT, considering the endocrine disrupting action of this compound?

 Response 1: The use of these concentration is justified by regulatory documents for human DDT exposure and differences in kinetics of DDT in the rats.

  1. Line 296: what do the authors mean by “intact female rats”?

Response 2:

Intact female and male rats were used to produce offspring for the control group. They were housed in the same conditions that the exposed rats but they did not consume DDT solution.

  1. Lines 303-304: this sentence is very confusing. What are the DDT consumption values for humans? Please explain this.

Response 3: The main source of DDT exposure in developed countries is food products. There are some regulations which determine maximal permissive concentrations of DDT in products of plant and animal origin. The main regulation in the Russian Federation is Technical Regulation of the Customs Union TR CU 021/2011 Concerning Safety of Food Products. 2015. We added the document to the reference list [53].

  1. When were the adrenal glands removed, at the 2 timepoints (42 and 70 days)? It is not very clear, maybe a scheme of the methods would be helpful.

 Response 4:

The half of the control and DDT-exposed rats were sacrificed on the 42nd day and the adrenals were removed, and the other rats on the 70th day of the experiment. We added the clarifications in the text.

  1. Line 325: why was embryonic tissue used as positive control?

Response 5:

The embryonic tissue was used because of higher expression of Oct4, Sox2 and Ascl1 in embryo than in adult organism.

  1. Lines 202-206: can you explain these differences, between the previous study and the present one?

Response 6:

We suppose that prenatal exposure to DDT shifts developmental rates of the adrenals during postnatal life and attenuates growth rates of the adrenals in pubertal period. The explanations are added to the text.

  1. Line 210: please explain and elaborate this sentence.

Response 7:

We have corrected and expanded this sentence.

  1. Line 353: how excessive can this growth be and what are the consequences? Please, further explore this issue.

Response 8:

We have added possible outcomes of excessive growth to the text.

  1. In the conclusion it should be explained how these results can contribute to new knowledge in the scientific area. How can the authors extrapolate these results to the human being? Compare DDT human exposure values to those used in the study.

Response 8:

We have expanded and supplemented the Conclusion. The information on correspondence of rat exposure levels to the humans are mentioned in the Materials and methods.

  1. Please propose a mechanism for the results and conclusions obtained in this manuscript (10.1016/j.mam.2021.101054, 10.1016/j.envpol.2023.121366 and 10.3389/fendo.2023.1059854).

Response 10:

The mechanisms for insufficient catecholamine production and regenerative potential are revealed in the present study. They include inhibition of tyrosine hydroxylase synthesis and impaired expression of transcription factor Ocrt4 and Shh ligand and activation of canonical Wnt.

Reviewer 3 Report

Comments and Suggestions for Authors

Review on the manuscript of Yaglova NV et al.: “Postnatal exposure to the endocrine disruptor DDT affects adrenomedullary chromaffin cells physiology and alters balance of mechanisms underlying cell renewal”.

In this study, Authors explored the influence of dichlorodiphenyltrichloroethane (DDT) on the function and renewal of chromaffin cells in adult adrenal medulla of male Wistar rats. The manuscript reports that exposed rats showed an increased proliferation and reduced expression of nuclear β-catenin. Therefore, these results suggest that DDT alters the physiology of adrenal gland.

Overall, I felt that this topic is of great interest, as millions of people are continuously exposed to pollutants, including DDT. Therefore, the elucidation of the risks posed by such substances may help in defining adequate measures to preserve or restore human health. The results are clearly presented, and the manuscript is well written, in general. In addition, I consider that the manuscript is precise on the question that Authors proposed to study. Thus, the issues that arise to me are listed below, so, I hope Authors find the following comments and suggestions useful.

1 - In the legend for figure 1, Authors indicate that “*” means p < 0.05, compared to the control of the appropriate age. However, in the graphs, the “*” is not represented. Can Authors clarify this situation?

2 - On postnatal day 42nd there was a downregulation of Ki-67 expression in DDT-exposed rats, whereas at postnatal day 70th DDT-exposed rats showed an upregulation of Ki-67 expression. Do Authors have any explanation for this dual effect of DDT? The same dual effect of DDT is still observed for Oct4 and Sonic Hedgehog expression. Therefore, I recommend Authors to explore these dual effects of DDT on these proteins in the discussion section.

3 - Authors indicate that immunohistochemical detection of transcription factor Sox2 did not reveal positive cells in the control and DDT-exposed rats during postnatal development. How do Authors know that Sox2 is not expressed? Did Authors used any positive control that could induce Sox2 expression? Without a positive control, it becomes difficult to take this conclusion. Therefore, if Authors want to keep these data in the manuscript, I recommend the use of a positive control. The same idea is valid for the Ascl1.

Comments on the Quality of English Language

Some minor mistakes detected.

Author Response

Dear reviewer! 

We are very grateful for all your comments and attention to the manuscript! The responses to your commenta are given below.

Comments and responses:

1 - In the legend for figure 1, Authors indicate that “*” means p < 0.05, compared to the control of the appropriate age. However, in the graphs, the “*” is not represented. Can Authors clarify this situation?

Response 1: There were no statistically significant differences between the control the exposed rats. We have removed the unnecessary caption from the legend to Fig.1.

2 - On postnatal day 42nd there was a downregulation of Ki-67 expression in DDT-exposed rats, whereas at postnatal day 70th DDT-exposed rats showed an upregulation of Ki-67 expression. Do Authors have any explanation for this dual effect of DDT? The same dual effect of DDT is still observed for Oct4 and Sonic Hedgehog expression. Therefore, I recommend Authors to explore these dual effects of DDT on these proteins in the discussion section.

Response 2:

We expanded the discussion and mentioned dual effects.

3 - Authors indicate that immunohistochemical detection of transcription factor Sox2 did not reveal positive cells in the control and DDT-exposed rats during postnatal development. How do Authors know that Sox2 is not expressed? Did Authors used any positive control that could induce Sox2 expression? Without a positive control, it becomes difficult to take this conclusion. Therefore, if Authors want to keep these data in the manuscript, I recommend the use of a positive control. The same idea is valid for the Ascl1.

Response 3:

All the antibodies were previously tested on embryonic tissue. The information is given in the Materials and methods.